# Orientation Probabilistic Movement Primitives on Riemannian Manifolds

**Leonel Rozo**[*1]     **Vedant Dave**[*1, 2]

[1]Bosch Center for Artificial Intelligence. Renningen, Germany.
[2]University of Leoben. Leoben, Austria.
leonel.rozo@de.bosch.com     vedant.dave@unileoben.ac.at

**Abstract:** Learning complex robot motions necessarily demands to have models that are able to encode and retrieve full-pose trajectories when tasks are defined in operational spaces. Probabilistic movement primitives (ProMPs) stand out as a principled approach that models trajectory distributions learned from demonstrations. ProMPs allow for trajectory modulation and blending to achieve better generalization to novel situations. However, when ProMPs are employed in operational space, their original formulation does not directly apply to full-pose movements including rotational trajectories described by quaternions. This paper proposes a Riemannian formulation of ProMPs that enables encoding and retrieving of quaternion trajectories. Our method builds on Riemannian manifold theory, and exploits multilinear geodesic regression for estimating the ProMPs parameters. This novel approach makes ProMPs a suitable model for learning complex full-pose robot motion patterns. Riemannian ProMPs are tested on toy examples to illustrate their workflow, and on real learning-from-demonstration experiments.

**Keywords:** Learning from Demonstration, Riemannian Manifolds

## 1   Introduction

For robots to perform autonomously in unstructured environments, they need to learn to interact with their surroundings. To do so, robots may rely on a library of skills to execute simple motions or perform complicated tasks as a composition of several skills [1]. A well-established way to learn motion skills is via human examples, a.k.a. learning from demonstrations (LfD) [2]. This entails a human expert showing once or several times a specific motion to be imitated by a robot. In this context, several models to encode demonstrations and synthesize motions have been proposed: Dynamical-systems approaches [3], probabilistic methods [4, 5], and lately, neural networks [6, 7].

Probabilistic movement primitives (ProMPs) [4], Stable Estimators of Dynamical Systems (SEDS) [8], Task-Parameterized Gaussian Mixture Models (TP-GMMs) [5], Kernelized MPs (KMPs) [9] and Conditional Neural MPs (CNMPs) [6] are some of the recent probabilistic models to represent motion primitives (MPs). While some of them were originally proposed to learn joint space motions (e.g., ProMPs), others have mainly focused on MPs in task space (e.g., TP-GMM), assuming that certain tasks, such as manipulation, may be more easily represented in Cartesian coordinates. The latter approach comes with an additional challenge, namely, encoding and synthesizing end-effector orientation trajectories. This problem is often overlooked, but the need of robots performing complicated tasks makes it imperative to extend MPs models to handle orientation data.

Orientation representation in robotics comes in different ways such as Euler angles, rotation matrices and quaternions. Euler angles are a minimal and intuitive representation, which however is not unique. They are also known to be undesirable in feedback control due to singularities [10, 11]. Rotation matrices are often impractical due to their number of parameters. In contrast, quaternions are a nearly-minimal representation and provide strong stability guarantees in close-loop orientation control [10]. Despite their antipodality (i.e., each rotation is represented by two antipodal points on the sphere $\mathcal{S}^3$), quaternions have gained interest in robot learning, control, and optimization due to their favorable properties [10, 12, 13, 14]. Pastor et al. [12] and Ude et al. [15] pioneered works

---

[*]Equal contribution.
5th Conference on Robot Learning (CoRL 2021), London, UK.

to learn quaternion MPs, where the classic DMP [3] is formulated to encode quaternion trajectories for robot motion generation. Its stability guarantees were later improved in [16], while Saveriano et al. [17] used this new formulation into a single DMP framework to learn full-pose end-effector trajectories. However, DMPs do not encode the demonstrations variability, which is often exploited to retrieve collision-free trajectories or to design robot controllers [4, 6].

Silvério et al. [13] proposed the first extension of TP-GMM for learning quaternion trajectories, which relies on unit-norm approximations. This was resolved in [18], where a Riemannian manifold formulation for both motion learning and reproduction was introduced. Despite TP-GMM offers good extrapolation capabilities due to its task-parameterized formulation, none of the foregoing works provides via-point modulation or trajectories blending. These issues were addressed by KMP [9] for position data, whose model builds on a previously-learned probabilistic MP, such as TP-GMM. KMP was then extended to handle orientation data [19], via a projection onto an Euclidean space using a fixed quaternion reference, i.e. a single Euclidean tangent space. Thus, model learning, via-point adaptation, and skill reproduction take place in Euclidean space, ignoring the intrinsic quaternions geometry. Note that such approximations lead to data and model distortions [18].

Although ProMPs have been used to learn Cartesian movements, their formulation does not handle quaternion trajectories. A possible solution would entail unit-norm approximations as in [13], but this approach fully ignores the geometry of the quaternions space and may lead to inaccurate models. An alternative and more sound solution relies on a Riemannian manifold formulation of ProMP, in the same spirit as [18]. However, two main difficulties arise: (*i*) learning of the model parameters does not accept a closed-form solution, and (*ii*) trajectory retrieval is constrained to lie on the sphere $\mathcal{S}^3$. We here provide solutions to these and related problems, which lead to the first ProMP framework that makes possible encoding and reproducing full-pose end-effector trajectories. In contrast to TP-GMM methods, our approach provides via-point modulation and blending capabilities, which are naturally inherited from the original ProMP. Unlike KMP, ProMP is a compact and standalone model, meaning that learning and reproduction do not rely on previously-learned MPs. Moreover, our approach is not prone to inaccuracies arising from geometry-unaware operations.

Specifically, we introduce a Riemannian manifold approach to learn orientation motion primitives using ProMP. Our extension builds on the classic ProMP [4] (summarized in § 2), and considers the space of quaternions as a Riemannian manifold. We propose to estimate the ProMP parameters using multivariate geodesic regression (see § 3), and we show how trajectory retrieval, modulation of the trajectory distributions, and MPs blending are all possible via a Riemannian probabilistic formulation. In § 4, we illustrate our approach on the unit-sphere manifold $\mathcal{S}^2$, and we learn realistic motion skills on a 7-DoF robotic manipulator featuring complex full-pose trajectories on $\mathbb{R}^3 \times \mathcal{S}^3$.

## 2 Background

### 2.1 ProMPs

Probabilistic Movement Primitives (ProMPs) [4] is a probabilistic framework for learning and synthesizing robot motion skills. ProMPs represent a trajectory distribution by a set of basis functions. Its probabilistic formulation enables movement modulation, parallel movement activation, and exploitation of variance information in robot control. Formally, a single movement trajectory is denoted by $\boldsymbol{\tau} = \{\boldsymbol{y}_t\}_{t=1}^T$, where $\boldsymbol{y}_t$ is a $d$-dimensional vector representing either a joint configuration or a Cartesian position at time step $t$ (additional time derivatives of $\boldsymbol{y}$ may also be considered). Each point of the trajectory $\boldsymbol{\tau}$ is represented as a linear basis function model

$$\boldsymbol{y}_t = \boldsymbol{\Psi}_t \boldsymbol{w} + \boldsymbol{\epsilon}_y \Rightarrow \mathcal{P}(\boldsymbol{y}_t | \boldsymbol{w}) = \mathcal{N}(\boldsymbol{y}_t | \boldsymbol{\Psi}_t \boldsymbol{w}, \boldsymbol{\Sigma}_{\boldsymbol{y}}), \tag{1}$$

where $\boldsymbol{w}$ is a $dN_\phi$-dimensional weight vector, $\boldsymbol{\Psi}_t$ is a fixed $d \times dN_\phi$-dimensional block diagonal matrix containing $N_\phi$ time-dependent Gaussian basis functions $\boldsymbol{\phi}_t$ for each DoF, and $\boldsymbol{\epsilon}_y \sim \mathcal{N}(\mathbf{0}, \boldsymbol{\Sigma}_{\boldsymbol{y}})$ is the zero mean i.i.d. Gaussian noise with uncertainty $\boldsymbol{\Sigma}_{\boldsymbol{y}}$ (see [4]). ProMPs employ a phase variable $z \in [0, 1]$ that decouples the demonstrations $\boldsymbol{\tau} = \{\boldsymbol{y}_t\}_{t=z_0}^{z_T}$ from the time instances, which in turn allows for temporal modulation. A table with the relevant notation is provided in Appendix 1.

ProMPs assume that each demonstration is characterized by a different weight vector $\boldsymbol{w}$, leading to a distribution $\mathcal{P}(\boldsymbol{w}; \boldsymbol{\theta}) = \mathcal{N}(\boldsymbol{w} | \boldsymbol{\mu}_{\boldsymbol{w}}, \boldsymbol{\Sigma}_{\boldsymbol{w}})$. Consequently, the distribution of $\boldsymbol{y}_t$, $\mathcal{P}(\boldsymbol{y}_t; \boldsymbol{\theta})$ is

$$\mathcal{P}(\boldsymbol{y}_t; \boldsymbol{\theta}) = \int \mathcal{N}(\boldsymbol{y}_t | \boldsymbol{\Psi}_t \boldsymbol{w}, \boldsymbol{\Sigma}_{\boldsymbol{y}}) \mathcal{N}(\boldsymbol{w} | \boldsymbol{\mu}_{\boldsymbol{w}}, \boldsymbol{\Sigma}_{\boldsymbol{w}}) d\boldsymbol{w} = \mathcal{N}(\boldsymbol{y}_t | \boldsymbol{\Psi}_t \boldsymbol{\mu}_{\boldsymbol{w}}, \boldsymbol{\Psi}_t \boldsymbol{\Sigma}_{\boldsymbol{w}} \boldsymbol{\Psi}_t^\mathsf{T} + \boldsymbol{\Sigma}_{\boldsymbol{y}}). \tag{2}$$

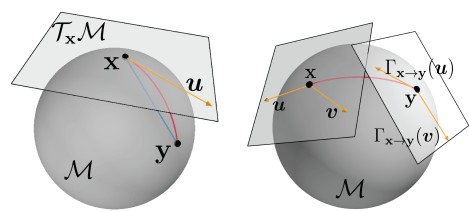

Figure 1: *Left*: Points on the surface of the sphere $\mathcal{S}^2$, such as $\mathbf{x}$ and $\mathbf{y}$ belong to the manifold. The shortest path between $\mathbf{x}$ and $\mathbf{y}$ is the geodesic (—), which differs from the Euclidean path (—). The vector $\boldsymbol{u}$ lies on the tangent space of $\mathbf{x}$ such that $\boldsymbol{u} = \text{Log}_{\mathbf{x}}(\mathbf{y})$. *Right*: $\Gamma_{\mathbf{x}\to\mathbf{y}}(\boldsymbol{u})$ and $\Gamma_{\mathbf{x}\to\mathbf{y}}(\boldsymbol{v})$ are the parallel transported vectors $\boldsymbol{u}$ and $\boldsymbol{v}$ from $\mathcal{T}_{\mathbf{x}}\mathcal{M}$ to $\mathcal{T}_{\mathbf{y}}\mathcal{M}$. The inner product between vectors is conserved by this operation.

**Learning from demonstrations:** The learning process of ProMPs mainly consists of estimating the weight distribution $\mathcal{P}(\boldsymbol{w}; \boldsymbol{\theta})$. To do so, a weight vector $\boldsymbol{w}_n$, representing the $n$-th demonstration as in (1), is estimated by maximum likelihood, leading to the solution of the form

$$\boldsymbol{w}_n = (\boldsymbol{\Psi}^\mathsf{T}\boldsymbol{\Psi} + \lambda\mathbf{I})^{-1}\boldsymbol{\Psi}^\mathsf{T}\boldsymbol{Y}_n, \tag{3}$$

where $\boldsymbol{Y}_n = \begin{bmatrix} \boldsymbol{y}_{n,1}^\mathsf{T} \dots \boldsymbol{y}_{n,T}^\mathsf{T} \end{bmatrix}^\mathsf{T}$ concatenates all the trajectory points, and $\boldsymbol{\Psi}$ consists of all the time instances for the basis-functions matrix $\boldsymbol{\Psi}_t$. Given a set of $N$ demonstrations, the weight distribution parameters $\boldsymbol{\theta} = \{\boldsymbol{\mu_w}, \boldsymbol{\Sigma_w}\}$ are estimated by maximum likelihood (see Algorithm 1 in Appendix 1).

**Trajectory modulation:** To adapt to new situations, ProMPs allow for trajectory modulation to via-points or target positions by conditioning the motion to reach a desired point $\boldsymbol{y}_t^*$ with associated covariance $\boldsymbol{\Sigma_y^*}$. This results into the conditional probability $\mathcal{P}(\boldsymbol{w}|\boldsymbol{y}_t^*) \propto \mathcal{N}(\boldsymbol{y}_t^*|\boldsymbol{\Psi}_t\boldsymbol{w}, \boldsymbol{\Sigma_y^*})\mathcal{N}(\boldsymbol{w}|\boldsymbol{\mu_w}, \boldsymbol{\Sigma_w})$, whose parameters can be computed as follows

$$\boldsymbol{\mu_w^*} = \boldsymbol{\Sigma_w^*}\left(\boldsymbol{\Psi}_t^\mathsf{T}\boldsymbol{\Sigma_y^*}^{-1}\boldsymbol{y}_t^* + \boldsymbol{\Sigma_w}^{-1}\boldsymbol{\mu_w}\right), \quad \boldsymbol{\Sigma_w^*} = \left(\boldsymbol{\Sigma_w}^{-1} + \boldsymbol{\Psi}_t^\mathsf{T}\boldsymbol{\Sigma_y^*}^{-1}\boldsymbol{\Psi}_t\right)^{-1}. \tag{4}$$

**Blending:** By computing a product of trajectory distributions, ProMPs blend different movement primitives into a single motion. The blended trajectory follows a distribution $\mathcal{P}(\boldsymbol{y}_t^+) = \prod_{s=1}^{S}\mathcal{P}_s(\boldsymbol{y}_t)^{\alpha_{t,s}}$, for a set of $S$ ProMPs, where $\mathcal{P}_s(\boldsymbol{y}_t) = \mathcal{N}(\boldsymbol{y}_t|\boldsymbol{\mu}_{t,s}, \boldsymbol{\Sigma}_{t,s})$, with associated blending weight $\alpha_{t,s} \in [0,1]$. The parameters of $\mathcal{P}(\boldsymbol{y}_t^+) = \mathcal{N}(\boldsymbol{y}_t^+|\boldsymbol{\mu}_t^+, \boldsymbol{\Sigma}_t^+)$ are estimated as follows

$$\boldsymbol{\Sigma}_t^+ = \left(\sum_{s=1}^{S}\alpha_{t,s}\boldsymbol{\Sigma}_{t,s}^{-1}\right)^{-1}, \quad \text{and} \quad \boldsymbol{\mu}_t^+ = \boldsymbol{\Sigma}_t^+\left(\sum_{s=1}^{S}\alpha_{t,s}\boldsymbol{\Sigma}_{t,s}^{-1}\boldsymbol{\mu}_{t,s}\right). \tag{5}$$

**Task parametrization:** ProMPs also exploit task parameters to adapt the robot motion to, for example, target objects for reaching tasks. Formally, ProMPs consider an external state $\hat{s}$ and learn an affine mapping from $\hat{s}$ to the mean weight vector $\boldsymbol{\mu_w}$, leading to the joint distribution $\mathcal{P}(\boldsymbol{w}, \hat{s}) = \mathcal{N}(\boldsymbol{w}|\boldsymbol{O}\hat{s} + \boldsymbol{o}, \boldsymbol{\Sigma_w})\mathcal{N}(\hat{s}|\boldsymbol{\mu_{\hat{s}}}, \boldsymbol{\Sigma_{\hat{s}}})$, where $\{\boldsymbol{O}, \boldsymbol{o}\}$ are learned using linear regression.

## 2.2 Riemannian manifolds

Since unit quaternions must satisfy a unit-norm constraint, they do not lie on a vector space, thus the use of traditional Euclidean methods for operating these variables is inadequate. We exploit Riemannian geometry to formulate ProMPs on quaternion space as presented in § 3. Formally, a Riemannian manifold $\mathcal{M}$ is a $m$-dimensional topological space with a globally defined differential structure, where each point locally resembles an Euclidean space $\mathbb{R}^m$. For each point $\mathbf{x} \in \mathcal{M}$, there exists a tangent space $\mathcal{T}_{\mathbf{x}}\mathcal{M}$ that is a vector space consisting of the tangent vectors of all the possible smooth curves passing through $\mathbf{x}$. A Riemannian manifold is equipped with a smoothly-varying positive definite inner product called a Riemannian metric, which permits to define curve lengths in $\mathcal{M}$. These curves, called geodesics, are the generalization of straight lines on the Euclidean space to Riemannian manifolds, as they are minimum-length curves between two points in $\mathcal{M}$ (see Fig. 1).

We exploit the Euclidean tangent spaces to operate with Riemannian data. To do so, we need mappings back and forth between $\mathcal{T}_{\mathbf{x}}\mathcal{M}$ and $\mathcal{M}$, which are the exponential and logarithmic maps. The exponential map $\text{Exp}_{\mathbf{x}} : \mathcal{T}_{\mathbf{x}}\mathcal{M} \to \mathcal{M}$ maps a point $\boldsymbol{u}$ in the tangent space of $\mathbf{x}$ to a point $\mathbf{y}$ on the manifold, so that it lies on the geodesic starting at $\mathbf{x}$ in the direction $\boldsymbol{u}$, and such that the

geodesic distance $d_{\mathcal{M}}$ between $\mathbf{x}$ and $\mathbf{y}$ equals the distance between $\mathbf{x}$ and $\mathbf{u}$. The inverse operation is the logarithmic map $\text{Log}_{\mathbf{x}} : \mathcal{M} \to \mathcal{T}_{\mathbf{x}}\mathcal{M}$. Another useful operation is the parallel transport $\Gamma_{\mathbf{x} \to \mathbf{y}} : \mathcal{T}_{\mathbf{x}}\mathcal{M} \to \mathcal{T}_{\mathbf{y}}\mathcal{M}$, which allows us to operate with manifold elements lying on different tangent spaces. The parallel transport moves elements between tangent spaces such that the inner product between two elements in the tangent space remains constant (see [20, 21] for further details). Finally, let us introduce a Riemannian Gaussian distribution of a random variable $\mathbf{x} \in \mathcal{M}$ as

$$\mathcal{N}_{\mathcal{M}}(\mathbf{x}|\boldsymbol{\mu}, \boldsymbol{\Sigma}) = \frac{1}{\sqrt{(2\pi)^d|\boldsymbol{\Sigma}|}} e^{-\frac{1}{2}\text{Log}_{\boldsymbol{\mu}}(\mathbf{x})^{\mathsf{T}} \boldsymbol{\Sigma}^{-1} \text{Log}_{\boldsymbol{\mu}}(\mathbf{x})}, \tag{6}$$

with mean $\boldsymbol{\mu} \in \mathcal{M}$, and covariance $\boldsymbol{\Sigma} \in \mathcal{T}_{\boldsymbol{\mu}}\mathcal{M}$. This Riemannian Gaussian corresponds to an approximated maximum-entropy distribution for Riemannian manifolds, as introduced by Pennec et al. [22]. Table 2 in Appendix 1 provides the different expressions for the Riemannian distance, exponential and logarithmic maps, and parallel transport operation for the sphere manifold $\mathcal{S}^m$.

## 2.3 Geodesic regression

A geodesic regression model generalizes linear regression to Riemannian manifolds. The regression is defined as $\mathbf{y} = \text{Exp}_{\tilde{\mathbf{y}}}(\boldsymbol{\epsilon})$, with $\tilde{\mathbf{y}} = \text{Exp}_{\mathbf{p}}(x\mathbf{u})$, where $\mathbf{y} \in \mathcal{M}$ and $x \in \mathbb{R}$ are respectively the output and input variables, $\mathbf{p} \in \mathcal{M}$ is a base point on the manifold, $\mathbf{u} \in \mathcal{T}_{\mathbf{p}}\mathcal{M}$ is a tangent vector at $\mathbf{p}$, and the error $\boldsymbol{\epsilon}$ is a random variable taking values in the tangent space at $\tilde{\mathbf{y}}$. As an analogy to linear regression, one can interpret $(\mathbf{p}, \mathbf{u})$ as an intercept $\mathbf{p}$ and a slope $\mathbf{u}$ (see [23] for details). Given a set of points $\{\mathbf{y}_1, \dots, \mathbf{y}_T\} \in \mathcal{M}$ and $\{x_1, \dots, x_T\} \in \mathbb{R}$, geodesic regression finds a geodesic curve $\gamma \in \mathcal{M}$ that best models the relationship between all the $T$ pairs $(x_i, \mathbf{y}_i)$. To do so, we minimize the sum-of-squared Riemannian distances (i.e., errors) between the model estimates and the observations, that is, $E(\mathbf{p}, \mathbf{u}) = \frac{1}{2} \sum_{i=1}^{T} d_{\mathcal{M}}(\hat{\mathbf{y}}_i, \mathbf{y}_i)^2$, where $\hat{\mathbf{y}}_i = \text{Exp}_{\mathbf{p}}(x_i\mathbf{u})$ is the model estimate on $\mathcal{M}$, $d_{\mathcal{M}}(\hat{\mathbf{y}}_i, \mathbf{y}_i) = \|\text{Log}_{\hat{\mathbf{y}}_i}(\mathbf{y}_i)\|$ is the Riemannian error, and the pair $(\mathbf{p}, \mathbf{u}) \in \mathcal{T}\mathcal{M}$ is an element of the tangent bundle $\mathcal{T}\mathcal{M}$. We can formulate a least-squares estimator of the geodesic model as a minimizer of such sum-of-squared Riemannian distances, i.e.,

$$(\hat{\mathbf{p}}, \hat{\mathbf{u}}) = \operatorname*{argmin}_{(\mathbf{p},\mathbf{u}) \in \mathcal{T}\mathcal{M}} \frac{1}{2} \sum_{i=1}^{T} d_{\mathcal{M}}(\hat{\mathbf{y}}_i, \mathbf{y}_i)^2. \tag{7}$$

The problem in (7) does not yield an analytical solution like (3). As explained by Fletcher [23], a solution can be obtained via gradient descent on Riemannian manifolds. Note that this geodesic model considers only a scalar independent variable $x \in \mathbb{R}$, meaning that the derivatives are obtained along a *single* geodesic curve parametrized by a *single* tangent vector $\mathbf{u}$. The extension to multivariate cases proposed by Kim et al. [24], where $\boldsymbol{x} \in \mathbb{R}^M$, requires a slightly different approach to identify multiple geodesic curves (viewed as "basis" vectors in Euclidean space). Multivariate general linear models on Riemannian manifolds (MGLM) [24] provides a solution to this problem. MLGM uses a geodesic basis $\boldsymbol{U} = [\boldsymbol{u}_1 \dots \boldsymbol{u}_M]$ formed by multiple tangent vectors $\boldsymbol{u}_m \in \mathcal{T}_{\mathbf{p}}\mathcal{M}$ of dimensionality $d = \dim(\mathcal{T}_{\mathbf{p}}\mathcal{M})$, one for each dimension of $\boldsymbol{x}$. Then, the problem (7) can be reformulated as

$$(\hat{\mathbf{p}}, \hat{\boldsymbol{u}}_m) = \operatorname*{argmin}_{(\mathbf{p},\boldsymbol{u}_m) \in \mathcal{T}\mathcal{M} \, \forall m} \frac{1}{2} \sum_{i=1}^{T} d_{\mathcal{M}}(\hat{\mathbf{y}}_i, \mathbf{y}_i)^2, \quad \text{with } \hat{\mathbf{y}}_i = \text{Exp}_{\mathbf{p}}(\boldsymbol{U}\boldsymbol{x}_i). \tag{8}$$

This multivariate framework allows us to compute the weight vector, analogous to (3), for a demonstration lying on a Riemannian manifold $\mathcal{M}$, for example $\mathcal{M} \equiv \mathcal{S}^3$.

## 3 Orientation ProMPs

When human demonstrations involve Cartesian motion patterns (via kinesthetic teaching or teleoperation), it is necessary to have a learning model that encapsulates both translation and rotation movements of the robot end-effector. This means that a demonstration trajectory $\boldsymbol{\tau} = \{\boldsymbol{y}_t\}_{t=1}^T$ is now composed of datapoints $\boldsymbol{y}_t \in \mathbb{R}^3 \times \mathcal{S}^3$, representing the full Cartesian pose of the end-effector at time step $t$. In this case, the challenge is learning a ProMP in the orientation space, as the Euclidean case in $\mathbb{R}^3$ follows the classic ProMP introduced in § 2.1. Therefore, we focus on how to extend ProMP to learn trajectories on $\mathcal{M} = \mathcal{S}^3$. First of all, let us introduce an equivalent expression for $\hat{\boldsymbol{y}}_i$, in the MGLM framework, such that it resembles the linear basis-function model

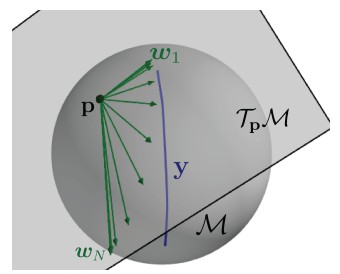

Figure 2: Illustration of multivariate general linear regression on the sphere manifold $\mathcal{S}^2$ used to learn the weights of orientation ProMPs. Given the trajectory $\mathbf{y}$, the origin $\mathbf{p}$ of the tangent space $\mathcal{T}_{\mathbf{p}}\mathcal{M}$, and the tangent weight vectors $\boldsymbol{w}_m$ are estimated via (12).

in (1). Specifically, the estimate $\hat{\mathbf{y}}_i = \mathrm{Exp}_{\mathbf{p}}(\boldsymbol{U}\boldsymbol{x}_i) \equiv \mathrm{Exp}_{\mathbf{p}}(\boldsymbol{X}_i\boldsymbol{u})$ with $\boldsymbol{U} \in \mathbb{R}^{d \times M}, \boldsymbol{x}_i \in \mathbb{R}^M$, $\boldsymbol{X}_i = \mathrm{blockdiag}(\boldsymbol{x}_1^{\mathsf{T}}, \ldots, \boldsymbol{x}_M^{\mathsf{T}}) \in \mathbb{R}^{d \times Md}$ and $\boldsymbol{u} = [\boldsymbol{u}_1^{\mathsf{T}} \ldots \boldsymbol{u}_M^{\mathsf{T}}]^{\mathsf{T}} \in \mathbb{R}^{Md}$. This equivalence proves useful when establishing analogies between the classic ProMPs formulation and our approach for orientation trajectories.

Similarly to (1), a point $\mathbf{y}_t \in \mathcal{M}$ of $\boldsymbol{\tau}$ can be represented as a *geodesic* basis-function model

$$\mathcal{P}(\mathbf{y}_t|\boldsymbol{w}) = \mathcal{N}_{\mathcal{M}}(\mathbf{y}_t|\mathrm{Exp}_{\mathbf{p}}(\boldsymbol{\Psi}_t\boldsymbol{w}), \boldsymbol{\Sigma}_{\mathbf{y}}), \quad \text{with} \quad \boldsymbol{\mu}_{\mathbf{y}} = \mathrm{Exp}_{\mathbf{p}}(\boldsymbol{\Psi}_t\boldsymbol{w}) \in \mathcal{M}, \tag{9}$$

where $\mathbf{p}$ is a fixed base point on $\mathcal{M}$, $\boldsymbol{w} = \left[\boldsymbol{w}_1^{\mathsf{T}} \ldots \boldsymbol{w}_{N_\phi}^{\mathsf{T}}\right]^{\mathsf{T}}$ is a large weight vector concatenating $N_\phi$ weight vectors $\boldsymbol{w}_n \in \mathcal{T}_{\mathbf{p}}\mathcal{M}$, $\boldsymbol{\Psi}_t$ is the same matrix of time-dependent basis functions as in (1), and $\boldsymbol{\Sigma}_{\mathbf{y}}$ is a covariance matrix encoding the uncertainty on $\mathcal{T}_{\boldsymbol{\mu}_{\mathbf{y}}}\mathcal{M}$. Two specific aspects about this formulation deserve special attention: (*i*) the mean $\boldsymbol{\mu}_{\mathbf{y}}$ of the Riemannian Gaussian distribution in (9), exploits the aforementioned equivalent formulation of MGLM; and (*ii*) the weight vectors forming $\boldsymbol{w}$ in (9) correspond to the vector composing the geodesic basis of MGLM.

As a demonstration is characterized by a different weight vector $\boldsymbol{w}$, again we can compute a distribution $\mathcal{P}(\boldsymbol{w}; \boldsymbol{\theta}) = \mathcal{N}(\boldsymbol{w}|\boldsymbol{\mu}_{\boldsymbol{w}}, \boldsymbol{\Sigma}_{\boldsymbol{w}})$. Therefore, the marginal distribution of $\mathbf{y}_t$ can be defined as

$$\mathcal{P}(\mathbf{y}; \boldsymbol{\theta}) = \int \mathcal{N}_{\mathcal{M}}(\mathbf{y}|\mathrm{Exp}_{\mathbf{p}}(\boldsymbol{\Psi}\boldsymbol{w}), \boldsymbol{\Sigma}_{\mathbf{y}})\mathcal{N}(\boldsymbol{w}|\boldsymbol{\mu}_{\boldsymbol{w}}, \boldsymbol{\Sigma}_{\boldsymbol{w}})d\boldsymbol{w}, \quad \text{with} \quad \boldsymbol{\mu}_{\mathbf{y}} = \mathrm{Exp}_{\mathbf{p}}(\boldsymbol{\Psi}\boldsymbol{w}) \in \mathcal{M}, \tag{10}$$

which depends on two probability distributions that lie on different manifolds. Using the Riemannian operations described in § 2, we obtain the final marginal (see Appendix 2.1)

$$\mathcal{P}(\mathbf{y}; \boldsymbol{\theta}) = \mathcal{N}_{\mathcal{M}}(\mathbf{y}|\hat{\boldsymbol{\mu}}_{\mathbf{y}}, \hat{\boldsymbol{\Sigma}}_{\mathbf{y}}), \quad \text{with} \quad \hat{\boldsymbol{\mu}}_{\mathbf{y}} = \mathrm{Exp}_{\mathbf{p}}(\boldsymbol{\Psi}\boldsymbol{\mu}_{\boldsymbol{w}}), \hat{\boldsymbol{\Sigma}}_{\mathbf{y}} = \Gamma_{\mathbf{p} \to \hat{\boldsymbol{\mu}}_{\mathbf{y}}}(\boldsymbol{\Psi}\boldsymbol{\Sigma}_{\boldsymbol{w}}\boldsymbol{\Psi}^{\mathsf{T}} + \tilde{\boldsymbol{\Sigma}}_{\mathbf{y}}), \tag{11}$$

where $\tilde{\boldsymbol{\Sigma}}_{\mathbf{y}}$ and $\hat{\boldsymbol{\Sigma}}_{\mathbf{y}}$ are parallel-transported covariances of the geodesic model (9) and the final marginal, respectively.

**Learning from demonstrations via MGLM:** For each demonstration $n$, we estimate a weight vector $\hat{\boldsymbol{w}}_n = \left[\hat{\boldsymbol{w}}_1^{\mathsf{T}} \ldots \hat{\boldsymbol{w}}_{N_\phi}^{\mathsf{T}}\right]^{\mathsf{T}}$ using MGLM (illustrated in Fig. 2). Firstly, we resort to the equivalent expression for $\mathbf{y}_t$ introduced previously, where $\mathrm{Exp}_{\mathbf{p}}(\boldsymbol{W}\boldsymbol{\phi}_t) \equiv \mathrm{Exp}_{\mathbf{p}}(\boldsymbol{\Psi}_t\boldsymbol{w})$, with $\boldsymbol{W} = \left[\boldsymbol{w}_1 \ldots \boldsymbol{w}_{N_\phi}\right]$ and $N_\phi$ being the number of basis functions. Secondly, we consider a demonstrated quaternion trajectory $\boldsymbol{\tau}_n = \{\mathbf{y}_t\}_{t=1}^T$ with $\mathbf{y}_t \in \mathcal{S}^3$. Then, analogous to (3) in Euclidean space, $\hat{\boldsymbol{w}}_n$ is estimated by leveraging (8), leading to

$$(\hat{\mathbf{p}}, \hat{\boldsymbol{w}}_m) = \underset{(\mathbf{p}, \boldsymbol{w}_m) \in \mathcal{T}\mathcal{M} \, \forall m}{\mathrm{argmin}} E(\mathbf{p}, \boldsymbol{w}_m), \quad \text{with} \quad E(\mathbf{p}, \boldsymbol{w}_m) = \frac{1}{2}\sum_{t=1}^{T} d_{\mathcal{M}}(\mathrm{Exp}_{\mathbf{p}}(\boldsymbol{W}\boldsymbol{\phi}_t), \mathbf{y}_t)^2, \tag{12}$$

where $\boldsymbol{\phi}_t$ is the vector of Gaussian basis functions at time $t$, and $\boldsymbol{W}$ contains the set of estimated tangent weight vectors $\hat{\boldsymbol{w}}_m \in \mathcal{T}_{\hat{\mathbf{p}}}\mathcal{M}$ (i.e., $N_\phi$ tangent vectors emerging out from the point $\mathbf{p} \in \mathcal{M}$). To solve (12), we need to compute the gradients of $E(\mathbf{p}, \boldsymbol{w}_m)$ with respect to $\mathbf{p}$ and each $\boldsymbol{w}_m$. As explained in Appendix 2.2, these gradients depend on the so-called adjoint operators, which broadly speaking, bring each error term $\mathrm{Log}_{\hat{\mathbf{y}}_t}(\mathbf{y}_t)$ from $\mathcal{T}_{\hat{\mathbf{y}}_t}\mathcal{M}$ to $\mathcal{T}_{\mathbf{p}}\mathcal{M}$, with $\hat{\mathbf{y}}_t = \mathrm{Exp}_{\mathbf{p}}(\boldsymbol{W}\boldsymbol{\phi}_t)$. Therefore, these adjoint operators can be approximated as parallel transport operations as proposed in [24]. This leads to the following reformulation of the error function of (12)

$$E(\mathbf{p}, \boldsymbol{w}_m) = \frac{1}{2}\sum_{t=1}^{T} \|\Gamma_{\hat{\mathbf{y}}_t \to \mathbf{p}}(\mathrm{Log}_{\hat{\mathbf{y}}_t}(\mathbf{y}_t))\|^2. \tag{13}$$

With the gradients of (13) (given in Appendix 2), we can now estimate both the vector $\mathbf{p}_n$ and the weight matrix $\boldsymbol{W}_n$, for each demonstration $n$. Note that each demonstration may lead to different estimates of $\mathbf{p}$, which defines the *base point* on $\mathcal{M}$ used to estimate $\boldsymbol{w}_m \in \mathcal{T}_{\mathbf{p}}\mathcal{M}$. This may produce different tangent spaces across demonstrations, and therefore diverse tangent weight vectors. An effective way to overcome this is to assume that all demonstrations share the same tangent space base $\mathbf{p}$, which is the same assumption made when defining the geodesic basis-function model (9). So, we only need to estimate $\mathbf{p}$ for a single demonstration, and use it to estimate all tangent weight vectors for the whole set of demonstrations. Then, given a set of $N$ demonstrations, the weight distribution parameters $\boldsymbol{\theta} = \{\boldsymbol{\mu}_{\boldsymbol{w}}, \boldsymbol{\Sigma}_{\boldsymbol{w}}\}$ are estimated by standard maximum likelihood as $\boldsymbol{w}_n \in \mathcal{T}_{\mathbf{p}}\mathcal{M} = \mathbb{R}^3 \subset \mathbb{R}^4$. The complete learning process for orientation ProMPs is given in Algorithm 2 of the supplementary material.

**Trajectory modulation:** We formulate a trajectory modulation technique (i.e., to adapt to new situations) by conditioning the motion to reach a desired trajectory point $\mathbf{y}_t^* \in \mathcal{M}$ with associated covariance $\boldsymbol{\Sigma}_{\mathbf{y}}^* \in \mathcal{T}_{\mathbf{y}_t^*}\mathcal{M}$. This results into the conditional probability $\mathcal{P}(\boldsymbol{w}|\mathbf{y}_t^*) \propto \mathcal{N}_{\mathcal{M}}(\mathbf{y}_t^*|\text{Exp}_{\mathbf{p}}(\boldsymbol{\Psi}_t \boldsymbol{w}), \boldsymbol{\Sigma}_{\mathbf{y}}^*)\mathcal{N}(\boldsymbol{w}|\boldsymbol{\mu}_{\boldsymbol{w}}, \boldsymbol{\Sigma}_{\boldsymbol{w}})$, which depends on two probability distributions that lie on different manifolds, similarly to (10). We leverage the fact that the mean $\boldsymbol{\mu}_{\mathbf{y}}$ depends on $\mathbf{p} \in \mathcal{M}$, which is the basis of $\mathcal{T}_{\mathbf{p}}\mathcal{M}$ where the weight distribution lies on. Thus, we rewrite the conditional distribution as follows

$$\mathcal{P}(\boldsymbol{w}|\text{Log}_{\mathbf{p}}(\mathbf{y}_t^*)) \propto \mathcal{N}(\text{Log}_{\mathbf{p}}(\mathbf{y}_t^*)|\boldsymbol{\Psi}_t \boldsymbol{w}, \tilde{\boldsymbol{\Sigma}}_{\mathbf{y}}^*)\mathcal{N}(\boldsymbol{w}|\boldsymbol{\mu}_{\boldsymbol{w}}, \boldsymbol{\Sigma}_{\boldsymbol{w}}) = \mathcal{N}(\boldsymbol{w}|\boldsymbol{\mu}_{\boldsymbol{w}}^*, \boldsymbol{\Sigma}_{\boldsymbol{w}}^*), \quad (14)$$

where $\tilde{\boldsymbol{\Sigma}}_{\mathbf{y}}^* = \Gamma_{\mathbf{y}_t^* \to \mathbf{p}}(\boldsymbol{\Sigma}_{\mathbf{y}}^*)$, and $\{\boldsymbol{\mu}_{\boldsymbol{w}}^*, \boldsymbol{\Sigma}_{\boldsymbol{w}}^*\}$ are the parameters to estimate for the resulting conditional distribution. Since both distributions now lie on $\mathcal{T}_{\mathbf{p}}\mathcal{M}$, which is embedded in the Euclidean space, we can estimate $\{\boldsymbol{\mu}_{\boldsymbol{w}}^*, \boldsymbol{\Sigma}_{\boldsymbol{w}}^*\}$ similarly to the classic ProMP conditioning procedure, with special care of parallel-transporting the covariance matrices. So, the updated parameters are

$$\boldsymbol{\mu}_{\boldsymbol{w}}^* = \boldsymbol{\Sigma}_{\boldsymbol{w}}^* \left( \boldsymbol{\Psi}_t^{\mathsf{T}} \tilde{\boldsymbol{\Sigma}}_{\mathbf{y}}^{*-1} \text{Log}_{\mathbf{p}}(\mathbf{y}_t^*) + \boldsymbol{\Sigma}_{\boldsymbol{w}}^{-1} \boldsymbol{\mu}_{\boldsymbol{w}} \right), \quad \text{and} \quad \boldsymbol{\Sigma}_{\boldsymbol{w}}^* = \left( \boldsymbol{\Sigma}_{\boldsymbol{w}}^{-1} + \boldsymbol{\Psi}_t^{\mathsf{T}} \tilde{\boldsymbol{\Sigma}}_{\mathbf{y}}^{*-1} \boldsymbol{\Psi}_t \right)^{-1}. \quad (15)$$

From the new weight distribution, we can obtain a new marginal distribution $\mathcal{P}(\mathbf{y}; \boldsymbol{\theta}^*)$ via (11).

**Blending:** When it comes to blend motion primitives in $\mathcal{M}$, one needs to consider that each of them is parametrized by a set of weight vectors lying on different tangent spaces $\mathcal{T}_{\mathbf{p}}\mathcal{M}$. Therefore, the weighted product of Gaussian distributions needs to be reformulated. To do so, we resort to the Gaussian product formulation on Riemannian manifolds introduced by Zeestraten [18], where the log-likelihood of the product is iteratively maximized using a gradient-based approach as proposed in [25]. We here provide the iterative updates of the blended distribution $\mathcal{P}(\mathbf{y}^+) = \mathcal{N}_{\mathcal{M}}(\mathbf{y}^+|\boldsymbol{\mu}^+, \boldsymbol{\Sigma}^+)$, while the full solution is given in Appendix 2.3. The iterative estimation of $\boldsymbol{\mu}^+$ is

$$\Delta_{\boldsymbol{\mu}_k^+} = \left( \sum_{s=1}^{S} \alpha_s \boldsymbol{\Lambda}_{\mathbf{y},s} \right)^{-1} \left( \sum_{s=1}^{S} \alpha_s \boldsymbol{\Lambda}_{\mathbf{y},s} \text{Log}_{\boldsymbol{\mu}_k^+}(\boldsymbol{\mu}_{\mathbf{y},s}) \right), \quad \text{and} \quad \boldsymbol{\mu}_{k+1}^+ \leftarrow \text{Exp}_{\boldsymbol{\mu}_k^+}(\Delta_{\boldsymbol{\mu}_k^+}), \quad (16)$$

for a set of $S$ skills, where $\boldsymbol{\Lambda}_{\mathbf{y},s} = \Gamma_{\boldsymbol{\mu}_{\mathbf{y},s} \to \boldsymbol{\mu}_k^+}(\boldsymbol{\Sigma}_{\mathbf{y},s}^{-1})$, and $\alpha_s$ is the blending weight associated to the skill $s$. After convergence at iteration $K$, we obtain the final parameters as follows

$$\boldsymbol{\mu}^+ \leftarrow \boldsymbol{\mu}_K^+ \quad \text{and} \quad \boldsymbol{\Sigma}^+ = \left( \sum_{s=1}^{S} \alpha_s \boldsymbol{\Lambda}_{\mathbf{y},s} \right)^{-1}. \quad (17)$$

**Task parametrization:** Classic ProMP allows for adapting the weight distribution $\mathcal{P}(\boldsymbol{w}; \boldsymbol{\theta}) = \mathcal{N}(\boldsymbol{w}|\boldsymbol{\mu}_{\boldsymbol{w}}, \boldsymbol{\Sigma}_{\boldsymbol{w}})$ as a function of an external task parameter $\hat{s}$, as explained in Section 2.1. This task parametrization straightforwardly applies to our method as the weight vectors $\boldsymbol{w}_n \in \mathcal{T}_{\mathbf{p}}\mathcal{M} \subset \mathbb{R}^4$, as long as the task parameter $\hat{s}$ is Euclidean. However, if $\hat{s} \in \mathcal{M}$, we can learn a joint probability distribution $\mathcal{P}(\boldsymbol{w}, \hat{s})$ using a Gaussian mixture model on Riemannian manifolds as proposed in [18, 26]. Subsequently, we can employ Gaussian mixture regression to compute $\mathcal{P}(\boldsymbol{w}|\hat{s}^*)$ during reproduction when a new task parameter $\hat{s}^*$ is provided. We refer the reader to the works of Zeestraten [18] and Jaquier et al. [26] for details on how to compute the distributions $\mathcal{P}(\boldsymbol{w}, \hat{s})$ and $\mathcal{P}(\boldsymbol{w}|\hat{s}^*)$.

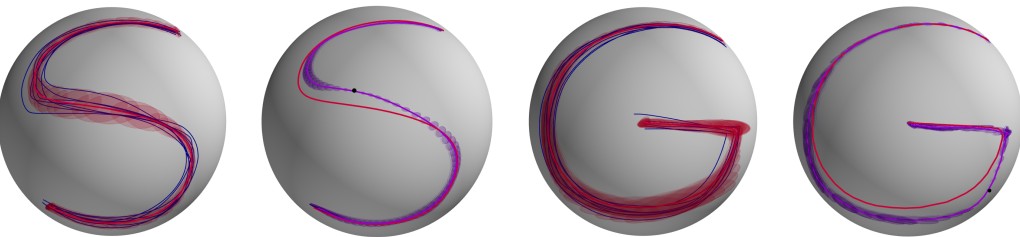

Figure 3: Demonstrated trajectories (—) in $\mathcal{S}^2$, marginal distribution $\mathcal{P}(\mathbf{y}; \boldsymbol{\theta})$ (mean —, covariance ⬤), and via-point adaptation (mean —, covariance ⬤, and via-point ●) for models trained over S and G datasets.

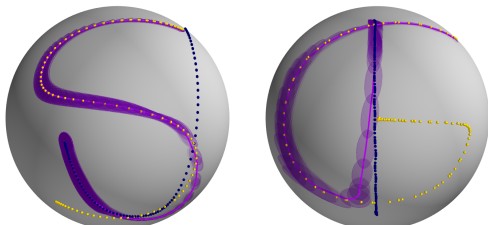

Figure 4: Trajectory distribution blending via orientation ProMP for datasets {S, J} and {G, I}. The resulting (blended) distribution trajectory (mean —, and covariance ⬤) starts from the first letter in the dataset (●) and then smoothly joins the trajectory of the second letter (●).

## 4 Experiments

**Synthetic data on $\mathcal{S}^2$:**    To illustrate how model learning, trajectory reproduction, via-point adaptation, and skills blending work in our approach, we used a dataset of hand-written letters. The original trajectories were generated in $\mathbb{R}^2$ and later projected to $\mathcal{S}^2$ by a unit-norm mapping. Each letter in the dataset was demonstrated $N = 8$ times, and a simple smoothing filter was applied for visualization purposes. We trained 4 ProMPs, one for each letter in the dataset {G, I, J, S}. All models were trained following Algorithm 2 with hyperparameters given in Appendix 3.1. Figure 3 shows the demonstration data, marginal distribution $\mathcal{P}(\mathbf{y}; \boldsymbol{\theta})$ computed via (10), and via-point adaptation obtained from (14) and (15), corresponding to the G and S models. The mean of $\mathcal{P}(\mathbf{y}; \boldsymbol{\theta})$ follows the demonstrations pattern, and the covariance profile captures the demonstrations variability in $\mathcal{S}^2$. Note that the trajectories G and S display very elaborated "motion" patterns that might be more complex than those observed in realistic robotic settings. Concerning the via-point adaptation, we chose a random point $\mathbf{y}^* \in \mathcal{S}^2$ with associated $\boldsymbol{\Sigma}_{\mathbf{y}}^* = 10^{-3}\boldsymbol{I}$ (i.e., high precision while passing through $\mathbf{y}^*$). As shown in Fig. 3, our approach smoothly adapts the trajectory and its covariance while accurately passing through $\mathbf{y}^*$.

To test the blending process, we used the following subsets of our original dataset: {G, I} and {S, J}. The goal was to generate a trajectory starting from the first letter in the set, and then smoothly switching midway to the second letter. Figure 4 shows the resulting blended trajectories for the two aforementioned cases, where our approach smoothly blends the two given trajectory distributions by following the procedure introduced in § 3. Note that the blending behavior strongly depends on the temporal evolution of the weights $\alpha_s \in [0, 1]$ associated to each skill $s$. We used a sigmoid-like function for the weights $\alpha_s^{(I)}$ and $\alpha_s^{(J)}$, while $\alpha_s^{(G)} = 1 - \alpha_s^{(I)}$ and $\alpha_s^{(S)} = 1 - \alpha_s^{(J)}$. The foregoing results show that our approach successfully learns and reproduces trajectory distributions on $\mathcal{S}^2$, and provides full via-point adaptation and blending capabilities. We now turn our attention to robotic experiments where we learn and synthesize full-pose movement primitives.

**Manipulation skills on $\mathbb{R}^3 \times \mathcal{S}^3$:**    To test our approach in a robotic setting, we consider a `re-orient` skill from [27], which involves lifting a previously-grasped object, rotating the end-effector, and placing the object back on its original location with a modified orientation (see Fig. 5). This skill features significant position and orientation changes, and it is therefore suitable to showcase the functionalities of our Riemannian ProMP. We collected 4 demonstrations of the `re-orient` skill via kinesthetic teaching on a Franka Emika Panda robot, where full-pose end-effector trajectories $\{\boldsymbol{p}_t\}_{t=1}^T$ were recorded, with $\boldsymbol{p}_t \in \mathbb{R}^3 \times \mathcal{S}^3$ being the end-effector pose at time step $t$. The data was used to train a ProMP on $\mathbb{R}^3 \times \mathcal{S}^3$, with position and orientation models learned using respectively a classic ProMP and our approach with hyperparameters given in Appendix 3.2.

Figure 6 shows the demonstrations (gray solid lines) and the mean of the marginal distribution $\mathcal{P}(\boldsymbol{p}; \boldsymbol{\theta})$ depicted as a black trajectory. Our model properly captures the motion pattern on $\mathbb{R}^3 \times \mathcal{S}^3$

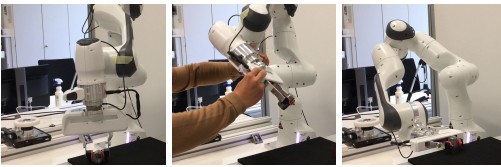

Figure 5: Snapshots of the human demonstrations of the `re-orient` skill [27]

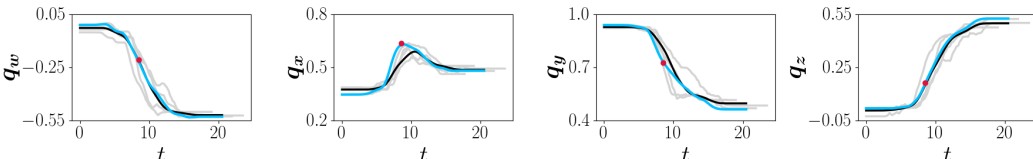

Figure 6: Time-series plot of the `re-orient` skill demonstrations (——), original mean trajectory (——) of the marginal distribution $\mathcal{P}(\boldsymbol{p};\boldsymbol{\theta})$, and resulting mean trajectory (——) of the new marginal distribution $\mathcal{P}(\boldsymbol{p};\boldsymbol{\theta}^*)$ passing through a given via-point (•) $\boldsymbol{p}^* \in \mathbb{R}^3 \times \mathcal{S}^3$. End-effector orientation represented as a quaternion $[\boldsymbol{q}_w, \boldsymbol{q}_x, \boldsymbol{q}_y, \boldsymbol{q}_z]$. Time axis given in sec.

for this skill. We then evaluated how this learned skill may adapt to a via point $\boldsymbol{p}^* \in \mathbb{R}^3 \times \mathcal{S}^3$, representing a new position and orientation of the end-effector at $t = 8.5\,\text{sec}$. By using the approach described in § 3, we computed a new $\mathcal{P}(\boldsymbol{p};\boldsymbol{\theta}^*)$, where the updated mean is required to pass through $\boldsymbol{p}^*$. Figure 6 displays the updated mean (light blue lines), which successfully adapts to pass through the given via-point. Note that the adapted trajectory exploits the variability of the demonstrations (i.e. the associated covariance) to adapt the trajectory smoothly. We also learned two classic ProMPs using an Euler-angle representation and a unit-norm approximation, using the same hyperparameters set. While both models retrieved a distribution $\mathcal{P}(\boldsymbol{p};\boldsymbol{\theta})$ similar to our approach, their performance is severely compromised in the via-point case, as they retrieve jerky trajectories with lower accuracy tracking w.r.t $\boldsymbol{p}^*$ (see Appendix 3.2 and supplemental simulation videos using PyRoboLearn [28]). These results confirm the importance of our Riemannian formulation for ProMP when learning and adapting full-pose end-effector skills. Appendix 3.2 reports learning and reproduction of two additional skills featuring motions of diverse complexity, demonstrating the versatility of our approach.

## 5 Discussion

Two main issues were identified in our approach: The effect of the hyperparameters on the trajectory distributions, and the increased computational cost of the weights estimation and blending process. The former problem is a limitation inherited from the classic ProMP, as the more complex the trajectory, the more basis functions we need. The smoothness of this encoding also relies on the basis functions width, which is often the same for all of them. We hypothesize that considering the trajectory curvature to distribute the basis functions accordingly and to define different widths may alleviate this hyperparameter tuning issue. The second problem arises as our weights estimation has no closed-form solution, and it relies on Riemannian optimization methods [29, 21] for solving the geodesic regression and for estimating mean vectors for distributions lying on $\mathcal{M}$. However, note that the weights are estimated only once and offline, meaning that the computations of $\mathcal{P}(\boldsymbol{p};\boldsymbol{\theta})$ or $\mathcal{P}(\boldsymbol{p};\boldsymbol{\theta}^*)$ are not compromised. Finally, the gradient-based approach to blend several ProMPs often converges in less than 10 iterations, making it a fairly fast approach for online blending settings.

As mentioned in Section 1, several methods overlook the problem of encoding orientation trajectories, or propose approximated solutions that produce distortion and inaccuracy issues (as shown in Appendix 3.2). Our Riemannian approach explicitly considers the geometric constraints of quaternion data, and therefore it represents a potential alternative to the Riemannian TP-GMM [18]. Nevertheless, the benefits of ProMP when compared to TP-GMM lie on the trajectory modulation and blending features, which endow a robot with a larger set of adaptation capabilities. It may be worth investigating how these formal methods dealing with full-pose trajectories compare to each other in terms of accuracy, adaptation, and extrapolation capabilities. In this context, benchmark works similar to [30] may bring important insights to the field. On a different note, the classic ProMP [4] also includes time derivatives of the considered variable, which was not covered in this paper. However, this extension is straightforward: This involves to include linear and angular velocities of the end-effector into the trajectory variables. Given that these are Euclidean variables, the main challenge arises when quaternions are part of the trajectory data.

**Acknowledgments**

The authors thank Noémie Jaquier (KIT) and Andras Kupcsik (BCAI) for their support and useful feedback on this work. We would also like to thank the reviewers and area chair for their highly useful recommendations.

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
