# OpenReview forum: "Orientation Probabilistic Movement Primitives on Riemannian Manifolds"
_robot-learning.org/CoRL/2021/Conference — CoRL2021 Poster_

### Official Review · Reviewer_G9zn · 2021-07-12

**Originality:** Fair
**Technical Quality:** Excellent
**Clarity Of Presentation:** Excellent
**Impact:** 3

**Recommendation:**

Weak Reject: I recommend rejecting the paper, but will not argue for my recommendation if the majority of other reviewers have a different opinion.

**Summary:**

This paper presents an extension of the probabilistic movements primitives framework for Riemannian Manifolds.
The authors propose a full ProMP framework adapted to the Riemannian Manifolds, which includes imitation learning, conditioning, blending, task parametrization.


**Issues:**

- Add a more complex robotic experiment to justify your claims and to show the capabilities of the robot.


**Reviewer Expertise:**

Very good: Comprehensive knowledge of the area

**Strengths And Weaknesses:**

The theory is well presented and well exposed. The formulation seems solid and quite easy to use.
However, the contribution is really incremental, as it consists mainly of combining existing techniques to enhance the ProMP framework.
The experimental evaluation is good, but a bit too simplistic, and it's not showing in a sufficiently convincing manner the advantage of considering the quaternions instead of euclidean variables to control robotic manipulator. While the authors prove that their method is better than plain ones, I would like to see more complex robotics experiments where orientation is fundamental e.g. peg-in-a-hole or other similar robotics tasks.

**Summary Of Recommendation:**

I think this is a borderline paper. The theoretical framework is solid, and the solved problem is really interesting. However the contribution is a bit incremental and the experimental evaluation, particularly on real robots is quite weak.
If I had the possibility, I would vote for borderline (nor weak accept, nor weak reject).
However, as I think that this paper could be improved, also in the rebuttal, I selected weak reject.

---

> ### Author Response · Authors · 2021-08-25
> **Response to reviewer G9zn**
>
> We would like to thank the reviewer for her/his time and the provided feedback which helped us improve our paper.
>
> **Additional experiments**:
> As suggested, we added new experiments showcasing successful learning and reproduction of full-pose trajectories for two robotic skills: *top-grasp* and *key-turning*. We believe that these experiments show that our Riemannian formulation is suitable to learn a wide range of motion skills featuring full-pose trajectories of varying complexity. Moreover, we added corresponding quantitative analysis for trajectory-based metrics (similarly to the *re-orient* skill).
>
> **Further comments**:
> We would like to emphasize that our extension of ProMP builds on a theoretically-grounded and rigorous formulation to learn trajectories lying on Riemannian manifolds, as pointed out by the other reviewers.
> We believe that the steps to make the original ProMP work on a Riemannian setting are non-trivial.
> Furthermore, our formulation provides the best quantitative results for different trajectory-based metrics.
> All in all, we think that our approach brings practical benefits (e.g., better trajectory-based metrics, no need to rely on unit-norm approximations or to handle Euler-angles non-uniqueness) while being theoretically sound.

---

### Official Review · Reviewer_MyGv · 2021-07-18

**Originality:** Very Good
**Technical Quality:** Very Good
**Clarity Of Presentation:** Very Good
**Impact:** 4

**Recommendation:**

Weak Accept: I recommend accepting the paper, but will not argue for my recommendation if the majority of other reviewers have a different opinion.

**Summary:**

The paper presents Riemannian formulation of Probabilistic Movement Primitive (ProMP) that enables the representation of quaternion trajectories which adheres to Riemannian geometry. In doing so, this paper extends ProMP which previously can only represent trajectories with Euclidean geometry, to those with Riemannian geometry, unlocking the possibility of encoding the full 6D pose trajectories and its uncertainty/probabilistic representation. The paper also explains how to perform Learning from Demonstration, trajectory modulation, blending, and task parametrization with this Riemannian ProMP.

**Issues:**

1. The explanation of Orientation ProMPs is sometimes unclear, or difficult to grasp/follow; sometimes I lose track during some algebraic manipulations. For example, on line 167, it would be helpful to more explicitly state what are the dimensions of the matrices U, xi, Xi, and u. Also, it would be very helpful (in maximizing the impact of the paper) to upload the source code implementation of the paper into the supplementary materials.

2. Moreover, the Log() and Exp() mapping in the supplementary materials are different from those in a referenced paper A. Ude, B. Nemec, T. Petric, and J. Morimoto: Orientation in Cartesian space dynamic movement primitives, as well as in a non-referenced paper J. Solà, J. Deray, and D. Atchuthan: A micro Lie theory for state estimation in robotics (https://arxiv.org/pdf/1812.01537.pdf).
Maybe it's useful to comment on what caused the differences.

3. Furthermore, naturally the dimension of the tangent space of 3D orientations (regardless of representation: quaternion, axis-times-angle, or rotation matrices) are 3; which makes sense because this tangent space is the same space as the space of angular velocities (omega vector in R^3). Can the author comment on why it is stated in line 205 that wn in tangent space TpM is a subset of R^4?

**Reviewer Expertise:**

Very good: Comprehensive knowledge of the area

**Strengths And Weaknesses:**

Strengths:
The paper explains the background on ProMP, Riemannian Manifold (tangent spaces, geodesic, exponential and logarithmic maps, and parallel transport), and geodesic regression very well.

Weaknesses:
The explanation of Orientation ProMPs is sometimes unclear, or difficult to grasp/follow; sometimes I lose track during some algebraic manipulations.

Moreover, the Log() and Exp() mapping in the supplementary materials are different from those in the referenced paper: A. Ude, B. Nemec, T. Petric, and J. Morimoto. Orientation in Cartesian space dynamic movement primitives.
No clarification is provided yet.

Furthermore, naturally the dimension of the tangent space of 3D orientations (regardless of representation: quaternion, axis-times-angle, or rotation matrices) are 3; which makes sense because this tangent space is the same space as the space of angular velocities (omega vector in R^3). However, it is stated in line 205 that wn in tangent space TpM is a subset of R^4. No clarification is provided yet.

See `Issues` for suggestions of improvements.

**Summary Of Recommendation:**

The paper contains novelty and deserves a publication, but can be improved by following the recommendations in the `Strengths and Weaknesses` or `Issues`.

---

> ### Author Response · Authors · 2021-08-25
> **Response to reviewer MyGv**
>
> We would like to thank the reviewer for the positive feedback and very useful comments to improve our paper. Below we answer each of the reviewer comments:
>
> **Notation improvements**:
> We added the dimensionality of the variables used in the intro of *Section 3*. They are now expressed as a function of the dimensionality of the tangent space $T_pM$, as introduced in *Section 2.3*. This keeps the notation general, but we believe that the introduced changes make the explanation clearer.
>
> **Exp() and Log() maps**:
> The difference of the logarithmic and exponential maps w.r.t the works cited by the reviewer is indeed a very good point to clarify. In few words, the differences come from how the retraction operation is defined and developed in Lie theory and Riemannian manifolds theory.
> In Lie theory, the exponential map is usually expressed at the identity element of the Lie group (requiring transport-like operations from and to the origin), while the exponential map used in the paper acts on any point $\mathbf{x} \in \mathcal{M}$. We added a more elaborated explanation in *Appendix 1.3* of the supplementary material, and added relevant references for the interested readers.
>
> **Tangent space dimensionality**:
> The reviewer is right, the tangent space $\mathcal{T}_p\mathcal{M}$ is actually $\mathbb{R}^3$. However, as the embedding space for the hypersphere $\mathcal{S}^3$ is the Euclidean space $\mathbb{R}^4$, we wanted to reflect this in the definition of the tangent space $\mathcal{T}_p\mathcal{M}$ (similarly to [1], Eq. (3.3)). This subtle detail is also brought up in the paper mentioned by the reviewer [2] (see Example 2 in page 3). We modified the definition to make this clearer.
>
> [1] Boumal, Nicolas. "An introduction to optimization on smooth manifolds." Available online, Aug (2020). http://sma.epfl.ch/~nboumal/book/index.html
>
> [2] J. Solà, J. Deray, and D. Atchuthan: A micro Lie theory for state estimation in robotics (https://arxiv.org/pdf/1812.01537.pdf

---

### Official Review · Reviewer_RYej · 2021-07-23

**Originality:** Very Good
**Technical Quality:** Excellent
**Clarity Of Presentation:** Excellent
**Impact:** 4

**Recommendation:**

Strong Accept: I recommend accepting the paper and will argue for my recommendation even if other reviewers hold a different opinion.

**Summary:**

This paper presents a principled extension of "probabilistic movement
primitives" or ProMPs from linear Gaussian models to data defined on
Riemannian manifolds. This is important for learning trajectories, since often
the trajectory of interest is specified in terms of orientations and poses,
which can be elegantly represented in terms of Riemannian manifolds. Consequently, the
extension of ProMPs to Riemannian manifolds is well-motivated, but involves a
nontrivial application of some fairly advanced machinery from Riemannian
geometry. The resulting approach is demonstrated on synthetic data and using a
real robot.

**Issues:**

## "Orientation" ProMPs

- The paper refers to "Orientation" ProMPs, but I don't see anything in the
  derivation in Section 3 that restricts this particular approach to
  orientations specifically - rather, this seems to be a general extension of
  ProMPs to arbitrary Riemannian manifolds (though admittedly, I don't have an
  immediate application to trajectory learning employing other manifolds that
  comes to mind). Might be worth a mention (maybe the authors have thoughts on
  this?)

## Discussion

- In the discussion, the issue of computational cost is raised, but I didn't see
  any quantitative mention about this (e.g. is there a noticeable increase in
  computation time). It's possible I missed something, but clarification here
  would be helpful.

## Minor Comments

- I recommend the excellent reference [1] for more detail on manifolds,
  especially for the concept of parallel transport. This might help a reader
  needing help with these concepts (it is a good supplement to the Absil,
  Mahony, and Sepulchre reference).

- Maybe more results could be provided (e.g. learning other skills than
  "re-orient"), but given the theoretical contribution this is minor.

[1] Boumal, Nicolas. "An introduction to optimization on smooth manifolds."
 Available online, Aug (2020). http://sma.epfl.ch/~nboumal/book/index.html

**Reviewer Expertise:**

Good: General knowledge of the area

**Strengths And Weaknesses:**

## Strengths

Overall, I felt this was an excellent paper.

- The paper is well-written and organized.

- The contribution is clear and relevant; the extension of probabilistic
movement primitives to Riemannian manifolds is well-motivated.

- The formulation is concise and easy to follow (not always easy to achieve when
  explaining Riemannian geometry)

- The method is demonstrated on synthetic data (which were really helpful for
  "visualizing" what was going on) and with real robot experiments.

- The animations in the supplemental video (e.g. adding via points) were very
  well done.

## Weaknesses

I have only minor comments for possible improvement of the paper, see Issues
below.

- The only "weakness" that stands out is the demonstration of only a single
skill with the real robot. It would be really neat to show more trajectories. As
mentioned below, though, the magnitude of the "theoretical" contribution makes
this a bit minor.

- I'd also suggest discussing a bit more about the timing / computational
efficiency. I'm particularly interested in the second issue raised in the
Discussion (regarding geodesic regression). How much more costly (e.g. in units
of time or relative order of magnitude) is this vs. the corresponding linear
system for ProMP?

**Summary Of Recommendation:**

I am recommending strong acceptance. This is a really nice paper. The proposed
work is well-motivated and clearly explained. Beyond that, results with
synthetic data and real robot experiments are provided. My only comments on
possible areas for improvement are minor.

---

> ### Author Response · Authors · 2021-08-25
> **Response to the reviewer RYej**
>
> We are glad to know that the reviewer enjoyed the paper and acknowledged the theoretical contributions that our paper introduces. Below we elaborate the changes made in the revised version of the paper.
>
> **Additional experiments**:
> As suggested, we added a couple of new experiments showcasing successful learning and reproduction of full-pose trajectories for *top-grasp* and *key-turning* skills. The results are reported in the updated supplementary material, where we added quantitative comparisons for several trajectory-based metrics similarly to what we reported previously for the *re-orient* skill.
>
> **Extension to other Riemannian manifolds**:
> Regarding the extension of our formulation to other types of manifolds, we fully agree with the reviewer, the formulation is not specifically designed for the hypersphere manifold, it actually extends to other Riemannian manifolds. For example, the robot joint space may be seen as a Torus manifold (i.e. a product of circle manifolds $S^1$), and therefore our formulation may be easily applied to learning skills in this space, while dealing with the geometry of this manifold (e.g., $0$ and $2*\pi$ represent the same joint position). Similar extensions may apply when learning impedance parameters profiles [1] as these belong to the manifold of symmetric positive definite matrices. However, as our motivation and experiments focus on orientation trajectories, we refer to our model as Orientation ProMPs in the title.
>
> **Computational cost**:
> We added quantitative results about the computational cost of our approach and compared them against the original ProMP formulation using both Euler angles and unit-norm approximations (see updated supplementary material). As it can be seen, the most computationally expensive part of our method corresponds to the multivariate geodesic regression used to compute the model weights. This process is fairly simple in the original ProMP formulation. However, as the weights computation is carried out offline, we do not see this as a potential issue of our approach. The trajectory retrieval, via-point conditioning and blending computations are not significantly compromised.
>
> [1] F. J. Abu-Dakka and M. Saveriano. "Variable Impedance Control and Learning—A Review". Front. Robot. AI. 2020.
>
> **Additional reference**:
> Finally, we also added N. Boumal's book as an additional reference for Riemannian manifolds, which is an excellent recommendation made by the reviewer.

---

### Official Review · Reviewer_qZSt · 2021-07-24

**Originality:** Very Good
**Technical Quality:** Excellent
**Clarity Of Presentation:** Very Good
**Impact:** 4

**Recommendation:**

Strong Accept: I recommend accepting the paper and will argue for my recommendation even if other reviewers hold a different opinion.

**Summary:**

This paper generalizes "Probabilistic Movement Primitives" from Euclidean spaces to Riemannian manifold.
The basic concepts in PMP (learning from demonstration, trajectory modulation, blending, task parametrization)
are mapped to the corresponding concept on the manifold. The mapping is done via geodesic regression, utilizing
the properties of smooth manifold. the method is demonstrated both on a synthetic data, and on a physical platform.
the simulations confirm the main claims of the paper.

**Issues:**

--

**Reviewer Expertise:**

Very good: Comprehensive knowledge of the area

**Strengths And Weaknesses:**

Strengths:
- the paper is well written, provides the required background
- the analogy between the PMP and the proposed method and the advantage is clear.
- the experiments are convincing.
- the videos in the supplementary are helpful and insightful.
- the limitations are reasonable. they are discussed, and the future steps are proposed.

Weakness:
- it would be worth to elaborate on the complexity of different complement of the method.
- adding more diverse experiment on the physical platform will be useful


**Summary Of Recommendation:**

I like this paper, because it has all the required: motivation, problem definition, rigour solution, validation.

---

> ### Author Response · Authors · 2021-08-25
> **Response to reviewer qZSt**
>
> We deeply appreciate the positive feedback about our work, and we would like to thank the reviewer for her/his encouraging comments.
>
> **Summary of updates**:
> As suggested, we evaluated our approach on a couple of additional experiments, reported in the *Appendix* of the updated *supplementary material*.
> The results show that our model successfully learns both the *top-grasp* and *key-turning* skills, demonstrating that our Riemannian formulation can be used to robustly learn different kinds of robot motion skills.
>
> We also added further clarifications and improved part of our notation to make our approach more understandable. We think these changes may also be of interest for this reviewer.

---

### Meta-Review · Area_Chair_Saj5 · 2021-08-14

**Recommendation:** Accept (Poster)
**Confidence:** 5

**Metareview:**

The reviewers agree that this is a very well written and thought through paper.
Most minor concerns regard some additional clarifications that should be included in the revised version.
The only major complaint is about the complexity of the provided experiments.
If possible some additional more complex real robot experiments would benefit the paper and make a stronger case for the approach.

## Post-Rebuttal Update:
The authors successfully addressed and clarified most of the raised issues and further improved the paper through the rebuttal phase.
Even if I would agree that the work is incremental, this would not a be a major concerns as long as the work is not trivial, which is not the case here.
I would ask the authors, however, to reference and discuss

[Gaussians on Riemannian Manifolds: Applications for Robot Learning and Adaptive Control](http://ras.papercept.net/images/temp/IROS/files/3131.pdf)  by Sylvain Calinon

in the camera ready version.

---

> ### Author Response · Authors · 2021-08-25
> **Response to Area Chair Saj5**
>
> We would like to thank the reviewers and the area chair for their useful feedback on the previous version of our paper.
>
> We have addressed the reviewers' suggestions and improved our paper accordingly. Text changes and new material are highlighted in blue in both the main paper and supplementary material.
>
> **Summary of updates**:
> Specifically, we added a couple of new robotic experiments to show how our approach can be used to learn a wide range of robot motion skills of varying complexity, as suggested by the reviewers and area chair.
> Moreover, we improved the model variables description in *Section 3*, and added clarifications regarding the exponential and logarithmic maps used in our paper, to make our approach more understandable.

---

### Decision · Program_Chairs · 2021-09-13

**Decision:**

Accept (Poster)

**Comment:**

The reviewers agree that this is a very well written and thought through paper.
Most minor concerns regard some additional clarifications that should be included in the revised version.
The only major complaint is about the complexity of the provided experiments.
If possible some additional more complex real robot experiments would benefit the paper and make a stronger case for the approach.

## Post-Rebuttal Update:
The authors successfully addressed and clarified most of the raised issues and further improved the paper through the rebuttal phase.
Even if I would agree that the work is incremental, this would not a be a major concerns as long as the work is not trivial, which is not the case here.
I would ask the authors, however, to reference and discuss

[Gaussians on Riemannian Manifolds: Applications for Robot Learning and Adaptive Control](http://ras.papercept.net/images/temp/IROS/files/3131.pdf)  by Sylvain Calinon

in the camera ready version.